# Detection of the Synthetic Cannabinoids AB-CHMINACA, ADB-CHMINACA, MDMB-CHMICA, and 5F-MDMB-PINACA in Biological Matrices: A Systematic Review

**DOI:** 10.3390/biology11050796

**Published:** 2022-05-23

**Authors:** Elisabet Navarro-Tapia, Jana Codina, Víctor José Villanueva-Blasco, Óscar García-Algar, Vicente Andreu-Fernández

**Affiliations:** 1Grup de Recerca Infancia i Entorn (GRIE), Institut D’Investigacions Biomèdiques August Pi i Sunyer (IDIBAPS), 08036 Barcelona, Spain; elisabet.navarro@campusviu.es (E.N.-T.); jana.codina.b@gmail.com (J.C.); ogarciaa@clinic.cat (Ó.G.-A.); 2Faculty of Health Sciences, Valencian International University (VIU), 46002 Valencia, Spain; vjvillanueva@universidadviu.com; 3Department of Neonatology, Hospital Clínic-Maternitat, ICGON, BCNatal, 08028 Barcelona, Spain

**Keywords:** synthetic cannabinoids, AB-CHMINACA, ADB-CHMNACA, MDMB-CHMICA, 5F-MDMB-PINACA, 5F-ADB, HRMS, toxicology, human matrix, detection

## Abstract

**Simple Summary:**

Synthetic cannabinoids were originally developed for scientific research and potential therapeutic agents. However, clandestine laboratories synthesize them and circumvent legal barriers by falsely marketing them as incense or herbal products. They have serious adverse effects, and new derivatives are continuously found in the market, making their detection difficult due to the lack of comparative standards. Human matrices are used to identify the type of synthetic cannabinoid and the time of its consumption. This review discusses the use of hair, oral fluid, blood, and urine in the detection and quantification of some of the major synthetic cannabinoids. Based on the results, some recommendations can be followed, for example, the use of hair to detect chronic and retrospective consumption (although sensitive to external contamination) and oral fluid or blood for the simultaneous detection of the parent compounds and their metabolites. If longer detection times than blood or oral fluid are needed, urine is the matrix of choice, although its pH may intervene in the analysis. This work highlights the use of new techniques, such as high-resolution mass spectrometry, to avoid the use of previous standards and to monitor new trends in the drug market.

**Abstract:**

New synthetic cannabinoids (SCs) are emerging rapidly and continuously. Biological matrices are key for their precise detection to link toxicity and symptoms to each compound and concentration and ascertain consumption trends. The objective of this study was to determine the best human biological matrices to detect the risk-assessed compounds provided by The European Monitoring Centre for Drugs and Drug Addiction: AB-CHMINACA, ADB-CHMNACA, MDMB-CHMICA, and 5F-MDMB-PINACA. We carried out a systematic review covering 2015 up to the present date, including original articles assessing detection in antemortem human biological matrices with detailed validation information of the technique. In oral fluid and blood, SC parent compounds were found in oral fluid and blood at low concentrations and usually with other substances; thus, the correlation between SCs concentrations and severity of symptoms could rarely be established. When hair is used as the biological matrix, there are difficulties in excluding passive contamination when evaluating chronic consumption. Detection of metabolites in urine is complex because it requires prior identification studies. LC-MS/MS assays were the most widely used approaches for the selective identification of SCs, although the lack of standard references and the need for revalidation with the continuous emergence of new SCs are limiting factors of this technique. A potential solution is high-resolution mass spectrometry screening, which allows for non-targeted detection and retrospective data interrogation.

## 1. Introduction

Synthetic cannabinoid receptor agonists are a group of substances designed as legal alternatives for cannabis that mimic the psychoactive effects of tetrahydrocannabinol (THC) by binding to cannabinoid receptors type 1 (CB1) and 2 (CB2). They are the largest group of new psychoactive substances (NPSs) monitored by the European Monitoring Centre for Drugs and Drug Addiction (EMCDDA). This organization defines an NPS as “a new narcotic or psychotropic drug, in pure form or in preparation, that is not controlled by the United Nations drug conventions, but which may pose a public health threat comparable to that posed by substances listed in these conventions” [1]. NPSs are continuously being designed, and thus, there is a high number of these types of substances in circulation. The most prolific year was 2015. In October 2020, the EMCDDA reported that at least 207 synthetic cannabinoids (SCs) had appeared in the drug market since 2008 [2].

SCs were originally developed as potential therapeutic agents [3]. However, clandestine laboratories started to produce these compounds and their homologs. JWH-018 was initially synthesized for research purposes but became the first SC detected as a “legal high” designed for inhalation [4]. Based on their structure, SCs are grouped into five main categories: classical cannabinoids (THC analogs), non-classical cannabinoids (including cyclohexylphenols), hybrid cannabinoids (include classical and non-classical cannabinoid structural features), aminoalkylindoles (the most common SC), and eicosanoids [5]. The chemical model for SCs proposed by the EMCDDA consists of a structure of 22 to 26 carbons with four key sections: the core and substituents, a link, a ring, and a tail [6]. A code can be given to each of these sections, facilitating the identification of the molecule without the need to sort out its complete chemical name (Figure 1).

Underground laboratories, mainly in Asian countries, clandestinely synthesize these substances as solids or oils in their pure form and then dissolve them, usually in ethanol or acetone. The mixture is then added to dried herbs such as lemon balm, mint, or thymus and sold in colorful sachets under various names (K2, Spice, Joker, Black Mamba, and so on). The most common mode of consumption is by smoking the dried material and, less frequently, by swallowing or brewing it as tea. They can also be purchased in liquid for vaporization in e-cigarettes [7]. To avoid regulatory obstacles, these products are labeled as “herbal incense” or “smoking mixtures”, “not for human consumption” [8]. This allows legal access to the drugs. They are distributed over the Internet and can be relatively easy to obtain. Many countries try to counteract this by banning substances that are under EMCDDA evaluation, but new analogs with minor changes in their structure, more affinity for the CB1 receptor, and that escape regulations are constantly appearing in the market [9].

SCs have a higher binding affinity for cannabinoid receptors than THC [10] and are usually fast-acting, reaching peak blood concentrations in less than 10 min when smoked [11]. Their shorter duration of action may increase the risk of dependency [12], and their effects are more pronounced than those of marijuana: elevated mood, relaxation, sensory perception changes, confusion, paranoia, and hallucinations, among others. Some adverse effects include tachycardia, agitation, vomiting, seizures, hyperglycemia, hypokalemia, stroke, myocardial infarction, acute kidney injury, and/or death [13,14]. Suicide attempts have also been described with the use of SCs due to the extreme anxiety these substances cause [15,16]. However, because of inter-batch differences—on occasions significant—the level of toxicity is often unknown and unpredictable. Many of these products contain SCs with unknown chemical composition, in higher doses than intended or in combination with either other NPS or with residues of the solvents used during the production process [17,18].

The rapid emergence of SCs, their chemical variety, and increasing number pose a challenge for their control and identification, as well as for their determination in biological specimens. Even if they bind to the same receptors, SCs have no structural similarity to THC, and the immunoassays designed to detect TCH-COOH—the main marijuana metabolite—are useless [19]. SCs go undetected in routine substance abuse testing at health centers; in cases of suspected use, specific tests must be asked for, with a clear advantage for the users.

Biological matrices may help detect drug intake and associate consumption to the clinical symptoms/signs and toxicity, although this is very challenging. On the one hand, the development and validation of analytical methods are always a step behind the appearance of new substances. On the other hand, screening methods must be very sensitive, as parent compounds are found in blood and oral fluid (OF) in low concentrations. Because of their rapid metabolism, detection windows are short, and other matrices, such as hair, are receiving more attention as they allow longer detection periods [20,21,22].

Analysis of SCs in biological matrices is mostly performed with gas or liquid chromatography combined with mass spectrometry (MS) or tandem mass spectrometry (MS/MS). Samples need to be prepared to reduce matrix effects and increase the sensibility, usually by liquid-liquid extraction (LLE) or solid-phase extraction (SPE) [23,24]. Non-targeted screening approaches, such as high-resolution mass spectrometry (HRMS) using quadruple time of flight (QTOF) instrumentation, are being developed for the detection of known and unknown compounds [25,26].

AB-CHMINACA, 5F-MDMB-PINACA, ADB-CHMINACA, and MDMB-CHMICA, are some of the new generations of cannabinoid compounds developed as marijuana substitutes and have been risk-assessed by the EMCDDA [27]. The first three have an indazole core, a common structural feature in many SCs, while MDMB-CHMICA is an indole-3-carboxamide derivative and the first SC receptor agonist to be risk-assessed by the EMCDDA [11]. Their characteristics and chemical structures differ (Table 1), but all are taken as recreational substances and are responsible for acute intoxications, deaths, and outbreaks of mass poisoning [28,29,30,31,32].

The prevalence of the use of these compounds in herbal smoking blends is unknown due to the variability and lack of information on the composition of these blends. Moreover, some parent compounds are metabolized to provide different phase I metabolites, hindering their identification by current analytical techniques [41]. The aim of this systematic review was to determine the best biological matrices for the determination of AB-CHMINACA, ADB-CHMINACA, MDMB-CHMICA, and 5F-MDMB-PINACA.

## 2. Materials and Methods

The preferred reporting items for systematic reviews and meta-analysis (PRISMA) statement was the methodology selected for the design of the present systematic review according to the guidelines of 2009 [42], as well as the update of the 2020 PRISMA statement [43]. The research team designed the definition of the research question and objectives, bibliographic search and data collection, evaluation, synthesis and comparison, critical appraisal of the analyzed scientific papers, as well as the presentation of the main findings and conclusions, showing the strengths and weaknesses of the studies (Figure 2). This review was registered in the PROSPERO database (CRD42022325490).

The aim of this study was to assess the detection of the synthetic cannabinoids AB-CHMINACA, ADB-CHMINACA, MDMB-CHMICA, and 5F-MDMB-PINACA in human biological matrices (OF, blood, urine, and hair) to establish the most appropriate technique for each compound. We ruled out a meta-analysis due to the differences in the techniques and parameters used for sample extraction and analysis, as well as the low number of studies for each biological matrix separately when considering the analyzed SC and technique applied, as this would lead to an important bias in the statistical results.

PubMed (https://pubmed.ncbi.nlm.nih.gov 1 April 2022), Cochrane Central Register of Controlled Trials (https://www.cochranelibrary.com/central 1 April 2022), and Scopus (www.scopus.com 1 April 2022) were the electronic databases consulted to collect the data using the following descriptors with the Boolean operators (AND/OR) in multiple combinations. In each database, we used the following terms for the search: ((AB-CHMINACA OR MDMB-CHMINACA OR 5F-MDMB-PINACA OR 5F-ADB OR ADB-CHMINACA OR MAB-CHMINACA) AND (hair OR blood OR serum OR plasma OR urine OR “oral fluid”)). Each compound was also searched individually for each biological matrix for better retrieval.

For the selection of the studies, the following inclusion criteria were defined: publications written in English aiming at detecting AB-CHMINACA, MDMB-CHMINACA, 5F-MDMB-PINACA, and ADB-CHMINACA, published between 1 January 2015, and March 2022 (no literature was found before 2015 due to the novelty of the compounds of interest). Four types of studies were excluded: (1) epidemiological studies and clinical cases, (2) studies in which the detection was performed in animals, herbal blends, or postmortem biological matrices, (3) studies on the metabolism and pharmacokinetics of SCs using in vitro approaches, animal models or in silico prediction, and (4) case reports with no validation information on the detection technique. Book chapters, correspondence, conference abstracts, and reviews were also excluded.

Original studies were first screened for title, and duplicates were removed. Next, abstracts were evaluated by applying the criteria for eligibility and data extraction of the studies meeting the inclusion criteria. Information extracted from each trial included: sample preparation, detection method, validation parameters, and usage on real samples. The quality of the studies was evaluated based on the limitations described in the articles and by assessing whether an application to real case samples and a description of validation parameters were included.

## 3. Results

We identified 215 articles in the database search, yielding 102 studies after the removal of duplicates. The latter were screened by title and abstract, and 35 met the inclusion criteria. The full text of these 35 studies was examined, and 11 were excluded. Following the electronic search, all the references from the selected articles were manually reviewed, and four articles extracted from these citations were included.

Twenty-eight articles aimed at detecting AB-CHMINACA, MBMD-CHMICA, 5F-MDMB-PINACA, ADB-CHMINACA, and their metabolites in human hair, OF, serum, plasma, and urine, were finally included in our analyses (Table 2); all were published between 2015 and 2022. One article described an immunoassay method validation, and the rest reported qualitative or quantitative chromatographic mass spectrometry assays (LC-MS/MS, GC-MS or LC-HRMS). The used validation parameters are presented in Table 3 and include the limit of detection (LOD), lower limit of quantification (LLOQ), linearity, bias or accuracy, precision, carryover, recovery, stability, interference, matrix effects and process efficiency, sensitivity, selectivity, and specificity. All of them are recommended parameters for method validation in forensic toxicology [44]. The largest number of the studies used blood samples, followed by studies using urine and OF.

### 3.1. Hair

Hair is a keratinized matrix that can allow detecting chronic drug abuse and complements urine or blood tests. It is a non-invasive matrix and can be easily stored. However, its pH variations due to dyes, the available quantity, or possible external contamination may hinder the analysis [67,68]. Four articles described method validation for SC detection in hair using LC-MS/MS. Cho and others developed a validated analytical method for the simultaneous determination of 19 SCs and 41 of their metabolites in human hair, which was applied to hair samples from individuals suspected to have used SCs [45]. AB-CHMINACA was the most frequently detected compound at concentrations ranging between 3.5 and 15,300 pg/mg. In most cases, its metabolite AB-CHMINACA M2 was also detected (0.5–35.1 pg/mg). The metabolites AB-CHMINACA M1A and M4 were only detected once. Sim and others analyzed 122 hair samples of SC abuse suspects by LC-MS/MS, showing similar results. The authors confirmed the presence of AB-CHMINACA in 37 samples in concentrations ranging from 2.2 to 1512 pg/mg. Likewise, AB-CHMINACA M2 was the most detected metabolite, with a ratio parent drug-metabolite of 18.8 and 151.8; AB-CHMINACA M4 was also detected in four samples [46]. Franz and others developed an LC-MS/MS method for the detection of SCs that was revalidated with the addition of new SCs. AB-CHMINACA, ADB-CHMINACA, MDMB-CHMICA, and 5F-MDMB-PINACA were detected in authentic samples at concentrations ranging between 0.8 and 1000 pg/mg, 3.5–210 pg/mg, 1.5–1300 pg/mg, and 1.3–2400 pg/mg, respectively [20]. In a previous study published in 2016, the same author detected AB-CHMINACA and AB-CHMINACA M2 in all hair segments by LC-MS/MS analysis using a sample from an individual with a known history of AB-CHMINACA consumption [47].

### 3.2. Oral Fluid

OF has several advantages over urine or blood; collection of this type of sample is non-invasive, and falsification is more difficult compared to urine, so this biological matrix has been widely considered in drug testing [69]. However, the disadvantages of using OF include the lack of knowledge regarding the pharmacokinetics of the tested agent in this matrix and the impossibility of obtaining postmortem samples. In addition, some SCs may be consumed by smoking, and oral contamination by the drug is possible if recently smoked [70]. That is, drug concentrations in saliva are much higher than in blood, providing inaccurate results. In turn, the probability of drug detection in these matrices increases. Four studies investigated SC detection in OF. Williams and others presented a method for quantifying AB-CHMINACA and/or other 18 SCs in OF using LC-MS/MS. The method was validated in authentic OF, but no confirmed positive samples were found when 12 OF samples submitted for routine testing were analyzed [48]. Cooman and others developed an LC-MS/MS method for detecting 24 NPS; they performed a blind verification study, detecting AB-CHMINACA in two samples (107.14 ng/mL and 59.73 ng/mL) [49]. One year later, Sorribes-Soriano and others described a procedure for detecting 5F-MDMB-PINACA and four other SCs in OF using GC-MS analysis following a semi-automated microextraction by packed sorbent (MEPS) extraction [50]. Although their validation data results showed high sensitivity, unfortunately, this method was not applied to authentic samples. Recently, gas chromatography coupled to a drift tube ion mobility spectrometer (GC-IMS) has also been tested to detect SCs in OF. Coupling GC with IMS is easier than combining it with mass spectrometry, and, in addition, both work at atmospheric pressure, not requiring vacuum pump systems. However, its narrow linear dynamic range makes quantitative analyses difficult. Denia and others first used this technique for the determination of illicit drugs, including the synthetic cannabinoid 5F-MDMB-PINACA [51]. The authors applied the technique to a synthetic OF sample containing the SC and obtained high precision in its identification, as well as a LOD and a limit of quantification (LOQ) of 30 and 90 μg/L, respectively. However, the technique was not applied to real case samples.

### 3.3. Blood, Serum, and Plasma

Fifteen articles described SC detection in hematic samples. Peterson and others reviewed the toxicological analysis of blood samples from suspected impaired driving cases with an LC-MS/MS method covering AB-CHMINACA. The authors reported a concentration range of AB-CHMINACA of 0.6–10 ng/mL and apparent severe impairment of the case histories even at low concentrations [52]. Later, Tynon and others developed a method for screening 34 SCs in whole blood and analyzed 1497 blood samples. They found that AB-CHMINACA (18.6%) and ADB-CHMINACA (15%) were the most frequently detected compounds (reporting a limit of 1 ng/mL and 0.1 ng/mL, respectively) [53]. LC-MS/MS blood sample analysis can also correlate acute ADB-CHMINACA intoxications with the severity of the symptoms. ADB-CHMINACA ranged from 1.3 to 14.6 ng/mL in the four analyzed samples, the concentrations depending on the estimated time between use and sample collection (6–9 h) [31]. A fatal case of MDMB-CHMICA poisoning determined by a validated LC-MS/MS method was also reported by the same author, in which antemortem blood concentration was 5.6 ng/mL [32]. Ninety-three different SCs, including AB-CHMINACA, ADB-CHMINACA, 5F-MDMB-PINACA, and MDMB-CHMICA, have also been detected by another validated LC-MS/MS method [54]. The authors applied the technique to 189 serum samples submitted for toxicological analysis and observed that 50% contained MDMB-CHMICA (range < LLOQ—8.7 ng/mL), 14% AB-CHMINACA (range < LLOQ—21.3), 2.13% 5F-MDMB-PINACA (range < LLOQ), and 2.13% ADB-CHMINACA (positive and 0.68 ng/mL) [54]. In some cases, positive findings could be linked to the described symptomatology, although no correlation between the intensity of psychoactive effects and compound concentration was found. This was also seen in a case series in which 26 blood samples from suspected SC users were analyzed by LC-MS/MS. Of the total number of samples, nine were positive for MDMB-CHMICA, often accompanied by other substances, although the authors were unable to show a correlation between clinical toxicity and MDMB-CHMICA concentration (range 5–22 ng/mL) [55]. One year later, Bäckberg and others described eight analytically confirmed MDMB-CHMICA intoxications in blood samples using a LC-MS/MS method (range: 3.4–86.4 ng/mL). However, this compound was not identified in urine samples. Most cases tested positive for other NPS [34].

HRMS has also been applied for toxicological detection in blood. In serum samples, 950 different compounds, including AB-CHMINACA (concentration of 1.2 μg/L) and MBMD-CHMICA (concentration range of 0.1–1.9 μg/L), were identified by liquid chromatography-quadrupole time-of-flight mass spectrometry (LC-QTOF-MS) [56]. Results were confirmed and quantified by a validated GC-MS, LC-QTOF-MS, proving to be superior by identifying 240% more drugs. The authors did not detect AB-CHMINACA nor MBMD-CHMICA by GC-MS in the samples. Later, the development of an LC/TOF-MS method by Saito and others allowed for determining AB-CHMINACA and another 46 SCs in illegal herbal products and blood [57], whereas Krotulski and others developed an LC-QTOF-MS method covering AB-CHMINACA, ADB-CHMINACA, 5F-MDMB-PINACA, and MDMB-CHMICA applied to 200 blood extracts [58]. With Krotulski’s non-targeted HRMS analytical approach, new SCs unreported in forensic toxicology, such as 5F-MDMB-PICA, 4-cyano CUMYL-BUTINACA, and 5F-EDMB-PINACA, were communicated. Moreover, compared to the original results, outcomes from 94% of the samples were in agreement with the original testing [58]. Ong and others developed an LC-MS/MS with a positive electrospray ionization (ESI) method for simultaneously detecting 29 SCs and their metabolites, as well as amphetamines and cannabinoids in 564 blood samples. Their results showed 132 positive blood samples for at least one SC. Among them, 5F-MDMB-PINACA was the second most prevalent analyte (53 samples) after AMB-FUBINACA (range of 1–100 ng/mL in 77% of the samples) [59]. LC-HRMS, LC-MS/MS, and LC-QTOF-MS are not the only methods to detect SCs in blood. Ultra-high pressure liquid chromatography quadrupole time-of-flight mass spectrometry (UHPLC-QTOF-MS) was used in 1314 postmortem blood samples. The aim of the work was to reprocess data files retrospectively and create a personal compound database and library (PCDL) to carry out a reevaluation for new findings [71]. In addition to synthetic opioids and designer benzodiazepines, SCs such as MDMB-CHMICA and AB-CHMINACA were analyzed. In the validation process, the authors detected relatively high concentrations of LOIs and low SC recoveries, so the risk of false negatives is more likely in this NPS group. Although the use of PCDL-containing diagnostic fragment ions was proved to be convenient in finding NPS retrospectively, the authors found no positive results regarding SCs [71].

### 3.4. Urine

Eleven studies assessed urine SC detection. Franz and others carried out a study to determine the diagnostic efficiency of two combined immunoassays (IA)—the “JWH-018 kit” and the “UR-144 kit” (Neogen, Lansing, MI, USA)—for detecting SC metabolites in 200 urine samples. Their results showed 2% sensitivity and 51% diagnostic accuracy, probably due to insufficient antibody cross-reactivity. Moreover, samples containing only metabolites of AB-CHMINACA, ADB-CHMINACA, or MDMB-CHMICA were not detected as positives [60]. It is very difficult to obtain precise measurements of 5F-MDMD-PINACA (5F-ADB) in urine samples due to pH variations (that intercede in the analysis of drugs of ionic nature) and the buffers used for extraction (that can result in the instability of the drug) [72]. Thus, the quick, easy, cheap, effective, rugged, and safe (QuEChERS) GC-MS/MS procedure was applied [35]. The results showed that the matrix was unaffected by the method and, most remarkably, drug recovery from alkaline urine was lower (27%) than natural (94%) or acidified urine (88%) with the same analyte concentration (5 ng/mL), probably because of partial degradation of the drug to its hydrolysis products. However, it was possible to identify the by-products of 5F-MDMB-PINACA hydrolysis in urine by LC-MS analysis after establishing their fragmentation pathways. Unfortunately, real case samples were not analyzed with this method. The influence of lipophilicity should also be considered when testing SCs in urine. Naphthoyl/benzoyl indole (Class I) and indole-3-carboxylate/carboxamide containing naphthol/quinol (class II) were undetectable in urine samples from 31 SC male users involved in car crashes in Japan [61]. Identification and quantitation of SCs by a validated LC/MS/MS procedure showed that indazole-3-carboxamide containing the valine/tert-leucine derivative (Class III)—the most hydrophilic SC—had a much lower log P range (2.29–3.81) than the other classes. The authors observed that 5F-MDMB-PINACA and AB-CHMINACA (Class III) SCs were easily excreted in the urine, as they are more soluble in water. Thus, Class III SCs were detected in 92.3% of the samples [61]. The detection of drugs in urine in drivers suspected to be Driving Under the Influence of Drugs (DUID) in Hungary showed a high prevalence of certain SCs, such as AB-CHMINACA and MDMB-CHMICA. Enzymatic hydrolysis and supported liquid extraction (SLE) were used for extraction and derivatization, whereas UHPLC-MS/MS was used for instrumental analysis [62]. After reviewing the data published in 2014 and 2015, the authors determined that the percentage of SC use was similar for the two years (15.1% and 19.0%, respectively, *p* > 0.05). The trend in SC use varied from year to year, being AB-CHMINACA the most popular in 2014 and MDMB-CHMICA in 2015. The most frequently detected SC in 2014 became the least frequently detected SC in 2015, presumably because of the inclusion of most of them as illicit drugs in Hungary at the beginning of 2015 [62]. However, the lack of information on the timing of sampling or lack of information in the registries might have led to an underestimation of the actual use of these drugs among this population.

Two articles included the development of in vivo and in vitro studies that aimed to identify the most relevant analytical targets beforehand. After investigating the metabolism of MDMB-CHMICA in vivo and in vitro using human liver microsomes (HLM), Franz and others included two of their main phase I metabolites in an LC-ESI-MS/MS routine SC screening method: M25 for its high selectivity and M30 for its high sensitivity to detect MDMB-CHMICA and structurally similar emerging SCs. M25 was the most abundant MDMB-CHMICA-specific metabolite to confirm consumption with authentic urine samples. The authors recommended that M25 be included in analytical methods for reliable differentiation of structurally related SCs, such as ADB-CHMICA and MDMB-CHMICA [63]. However, as stated by the authors, the protocol used for sample preparation did not allow the analysis of the complete urinary metabolic profile, so further works on the structure of some metabolites need to be carried out. Years later, the in vitro metabolism of 5F-MDMB-PINACA using pooled HLM and later authentication in 30 human urine samples was performed [64]. M20, M8, and M17 were the most sensitive and specific urinary markers and were therefore incorporated into an LC-HRMS screening method for determining 5F-MDMB-PINACA and 5F-MDMB-PINACA carboxylic acid (M20) levels and the corresponding blood samples. Concentration ranges in blood were 0.13–3.88 ng/mL for 5F-MDMB-PINACA and 0.23–67.09 ng/mL for M20. In urine, the parent compound was rarely detected (5.7 ng/mL was the highest level), whereas M20 was found in concentrations ranging from 0.16 to 99.92 ng/mL. The authors pointed out the utility and applicability of simulated metabolic reactions with HLM to screen for emerging SCs, as well as the sensitivity and specificity of M20, M8, and M17 to distinguish 5F-MDMB-PINACA consumption from the possible use of other tert-leucine derivatives [64]. Several quantitative screening methods in urine by LC-MS/MS have been published (Table 2). Moreover, UHPLC-QTOF-MS, which allows obtaining full-spectrum data and enables a retrospective analysis of previous data [71], was also validated for the quantification and confirmation of 35 SC metabolites from 1000 consecutive routine urinary samples [25]. Among those metabolites, AB-CHMINACA M1A, AB-CHMINACA 3-carboxyindazole, and AB-CHMINACA M2, generated from the intake of AB-CHMINACA, reached limits of confirmation of 10, 2.5, and 1 ng/mL, respectively, while their limits of quantification were 10, 0.25, and 1 ng/mL. At the clinical level, 2.3% of samples were positive (mostly metabolites derived from the intake of JWH-018). Although there was no detection of AB-CHMINACA intake, this could be due to a high matrix effect and thereby a limitation for the analytical quality of their metabolites. The authors suggest this method should be considered semi-quantitative for the analytes AB-CHMINACA M1A and AB-CHMINACA 3 carboxyindazole [25]. In addition, they recommend analyzing samples right after their processing, using glassware and a temperature of 10 °C to minimize the loss of analytes.

As reported in the above-mentioned study, it was not possible to quantify parent AB-CHMINACA in urine from suspected cases in an outbreak of AB-CHMINACA consumption in Florida [65]. The authors used LC/QTOFMS in plasma, serum, and urine samples. Concentrations in serum ranged between 0.4 and 13.8 ng/mL, and quantifiable amounts of M2 (from 39 to 303 ng/mL) were determined. In urine samples (*n* = 21), the predicted metabolites were found in around half of the samples, but none contained the parent compound. This outbreak study is noteworthy as it involved a multidisciplinary collaboration. Because of the novelty of AB-CHMINACA, no reference standards for its detection in the samples were available. The work group between laboratories, biotechnology companies, and governmental entities succeeded in identifying and seizure the product causing the outbreak. Some months later, AB-CHMINACA was placed on Schedule I of the Controlled Substances Act [73].

Other methods for detecting SCs are not based on the structure of these molecules. Cannaert and others [66] designed an alternative screening method considering the activity of the SC. They developed cannabinoid receptor activation assays with stable cell systems. The binding of SCs or their metabolites to the cannabinoid receptor restores the Nanoluciferase (NanoLuc) activity that can be easily detected by a bioluminescent signal. The method incorporated major phase I metabolites of AB-CHMINACA and ADB-CHMINACA, previously identified by LC-MS/MS analysis in urine samples. Cannabinoid receptor activation by SC and their metabolites were detected in subnanomolar concentrations, and the metabolites retained their activity at the cannabinoid receptors. However, the highest signals were detected with the parent compounds—AB-CHMINACA and ADB-CHMINACA—due to a decreased metabolization activity. This method allows the detection of SCs based on their activity, without the need to know their structure or the metabolites. Although this simplifies the detection of new unknown compounds, the authors recommend the use of an analytical procedure for confirmation [66].

## 4. Discussion

New SCs are constantly being developed, which represents a major challenge for analytical laboratories, often with no available analytical standards, hindering their identification in biological matrices. The use of highly specific and sensitive analytical methods is mandatory to detect SC consumption. In addition, the appropriate biological matrix is a key factor that should consider the purpose of the analysis, the ease of sampling, and the available instrumentation.

SC intake can be retrospectively detected by segmental hair analysis, with a detection window from months to years [74]. Hair samples have suitable stability and can be easily and non-invasively collected, preferably under supervision to prevent manipulation. The main purpose of hair analysis is to distinguish between external contamination and drug incorporation following consumption [74]. Franz and others [20] detected SCs at periods that did not correlate with the compounds’ availability in the “legal high” market. This was attributed to external contamination, such as handling of drug material or exposure to side stream smoke. In another study, the analysis of MDMB-CHMICA smoke condensate showed that the M30 metabolite could be formed pyrolytically under smoking conditions and condensate on the hair as a form of external contamination [63]. Although the analysis of metabolites has been proposed as a useful tool to exclude passive contamination [46], SC metabolites could also be generated ex vivo and detected in externally contaminated hair [47], as has been seen with other drugs such as cocaine [75]. Therefore, hair is useful for evaluating chronic SC consumption, but complementary analysis of other biological matrices may be needed to rule out passive contamination. We have shown the results of a keratin-based matrix hair and would like to point out that other similar matrices are being studied, such as fingernails or toenails [76]. The authors determined cannabinoid distribution (not SCs) in nails compared to hair, and analyte concentrations in fingernails were much higher than in toenails and hair. Therefore, although further validated studies on the use of these unconventional matrices based on keratin in the detection of SCs are needed, it may be a promising alternative when hair cannot be obtained. In fact, not only the consumption of SCs has been studied in these non-conventional matrices, but other drugs such as ecstasy, cocaine, or ketamine have also been assessed in these two matrices, giving significantly higher values in fingernails than in toenails [77].

Obtaining samples from OF is also easy and non-invasive and commonly used at workplaces and DUID testing. A drawback when using this matrix is the limited material available, as SCs are highly potent and thus found in very low concentrations with a typically described detection time of hours to days [50]. OF concentration generally indicates the free pharmacologically active component in serum—not bound to proteins—reflecting the concentration at the active site [74]. OF metabolite disposition has not yet been elucidated, but targeting metabolites using this matrix may help document active intake, as passive environmental SC exposure has been described to produce positive OF results [24]. All OF studies included in this review developed methods for the detection and/or quantification of solely parent compounds [48,49,50,51]. Moreover, none of the articles presented results from positive real case samples. Although new validated techniques have been developed for the detection of SCs in saliva, such as GC-IMS, which offer a sensitivity in the same range of conventional simple quadrupole detectors [51], positive confirmation in real cases has to be proved in order to ascertain its use.

Unlike saliva or hair, blood collection is an invasive procedure. The method needs to be sensitive due to the short window of detection of the drugs of abuse (1–2 days) and the short half-life and low concentration of the compounds [78]. The effects of SC drugs show the best correlation with hematic samples compared to other matrices; however, most studies included in this review could rarely correlate SC blood concentrations to the severity of the symptoms, sometimes because the cases tested positive for other NPS [34,54,55]. In addition, the lack of data such as the dose of the used SC or the time between taking the drug and the onset of symptoms makes this analysis difficult. In a remarkable work by Institóris and others with 116 suspected cases of drug abuse in the Budapest region [79], blood samples confirmed SCs as the most prevalent compound in 41.4% of the positive patients. 5F-MDMB-PINACA was the most frequently used compound, but no correlation between blood levels and symptom severity was found. Bradypsychia and confusion were the most frequent symptoms seen in 5F-MDMBPINACA users (when used alone), but they were not significantly different from those generated in individuals in which other SCs were detected, e.g., MDMB-CHMICA. All these studies support the need for prospective studies to determine the exact relationship between blood SC levels and the severity of the symptoms.

SC detection methods using blood mainly target parent compounds, as metabolites are more rarely found. However, in Krotulski’s HRMS approach, a high number of metabolites were identified, including 5F-MDMB-PINACA M20 [58]. In 2021, this compound was found to be more stable in blood under all storage conditions than the parent compound after LC–QTOF-MS analysis [80]. The analysis of blood extracts from a forensic toxicology laboratory showed negative results for 5F-MDMD-PINACA, but its metabolite 5F-ADB 3,3-dimethylbutanoic acid was detected [58]. Likewise, quantifiable AB-CHMINACA metabolites such as M2 were found in two plasma samples in Tyndall’s study even though the parent compound was at levels below the lower limit of quantification [65]. The presence of metabolites at higher levels than the parent SCs in the blood (also observed by Ong and others [59]) indicates that simultaneous analysis of metabolites and parent compounds in this biological matrix should be carried out to avoid discarding positive samples.

In comparison with blood, urine is obtained non-invasively in large volumes. In this matrix, SCs are metabolized rapidly and extensively, and parent compounds can rarely be found. Thus, the development of targeted screening methods requires knowing the metabolic profiles of the new emerging compounds. Phase I biotransformation reactions are mainly oxidation processes involving CYP450 enzymes: ester hydrolysis, hydroxylation, carboxylation, and defluorination of fluorinated analogs [24,41]. Conjugation by glucuronidation or sulfation is the major phase II reaction [41]. Phase I metabolites are more stable than phase II metabolites and respond better to mass spectrometers; thus, they are the best markers [24]. Therefore, the targets in urine should be metabolites, prior identification via in vitro studies to identify the metabolic products of the compound, which can later be included in a new analytical method. The lack of reference standards has prevented the detection of metabolites in urine samples [66], so further studies on the metabolic profile of these compounds are needed.

Studies have been conducted to elucidate the metabolites derived from SCs to help detect the uptake of these drugs. Dybowski and others [35] established a fragmentation pathway to elucidate generated metabolites from 5F-MDMB-PINACA, while Franz and others [63] investigated in vitro metabolism of MDMB-CHMICA. However, in both studies, the exact structure of some metabolites remained unknown as they were not elucidated by nuclear magnetic resonance spectroscopy. Similarly, the LC-MS/MS method described by Franz and others in 2017 could not be fully validated for all analytes covered for reference standards were not available [60]. Therefore, in urine, unlike blood, saliva, or hair, it is more common to find metabolites than parent compounds. Furthermore, it also does not correlate as well with plasma-free drug concentrations as, for example, saliva [74]. However, the detection time is longer compared to OF or blood [78]; even a case of an elimination phase of approximately 1 year has been detected in an extensive consumption profile [81].

Regarding the detection methods, immunoassay tests for the screening of certain SCs are commercially available but need confirmation by LC-MS/MS and have become rapidly obsolete, as shown by Franz and others [60]. The rapid development of new SCs is a limitation even for targeted mass spectrometry screening methods, which can only detect the analytes they were designed for, and thus require continuous updating and validation. Routine analytical approaches cannot monitor the new trends in the drug market because when a substance is finally identified and incorporated into an analytical panel, it is no longer used. New approaches, such as HRMS, although expensive, can tentatively identify unknown substances without the need for reference standards. Once the reference standards become available, retrospective analysis can be performed. Krotulski and others incorporated a non-targeted data mining technique called SWATH acquisition. This technique allows reprocessing samples to detect new compounds that had not been incorporated for testing when they first appeared on the market [58]. Another potential identification tool was described by Saito and others who studied the correlation between the logarithm of retention time and logPow (octanol/water partition coefficient); the authors concluded that logPow is useful for estimating the retention time of unknown SCs and may be of help for compound identification [57]. Scientific production regarding HRMS has markedly increased since 2015, and most likely, it will become the tool of excellence in toxicology laboratories. Non-targeted HRMS techniques seem to be the best choice to identify and quantify known and unknown SCs and metabolites with high sensitivity and selectivity. They allow retrospective data analysis and meet the demands of a field dealing with the continuous development of new compounds. The HRMS technique is expensive but can be of great use in cases when precise toxicological analyses are required (deaths due to intoxication, traffic accidents, or criminal offenses), although it is not the only valid one.

In this review, there may be a certain reporting bias, as studies that did not detail validation parameters or those that focused exclusively on postmortem analysis were excluded. In addition, due to the novelty of the compounds, little information has been available, while the findings presented here may soon become obsolete as new SCs appear in the market. Although we cannot infer the results to the whole set of current SCs, we hope researchers and clinicians can guide their SC detection approaches based on some key findings and limitations compiled in this review.

## Figures and Tables

**Figure 1 biology-11-00796-f001:**
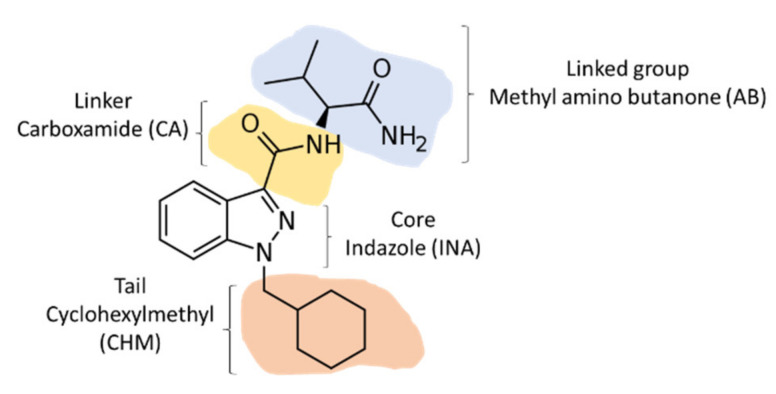
Structure of the synthetic cannabinoid AB-CHMINACA based on the EMCDDA model.

**Figure 2 biology-11-00796-f002:**
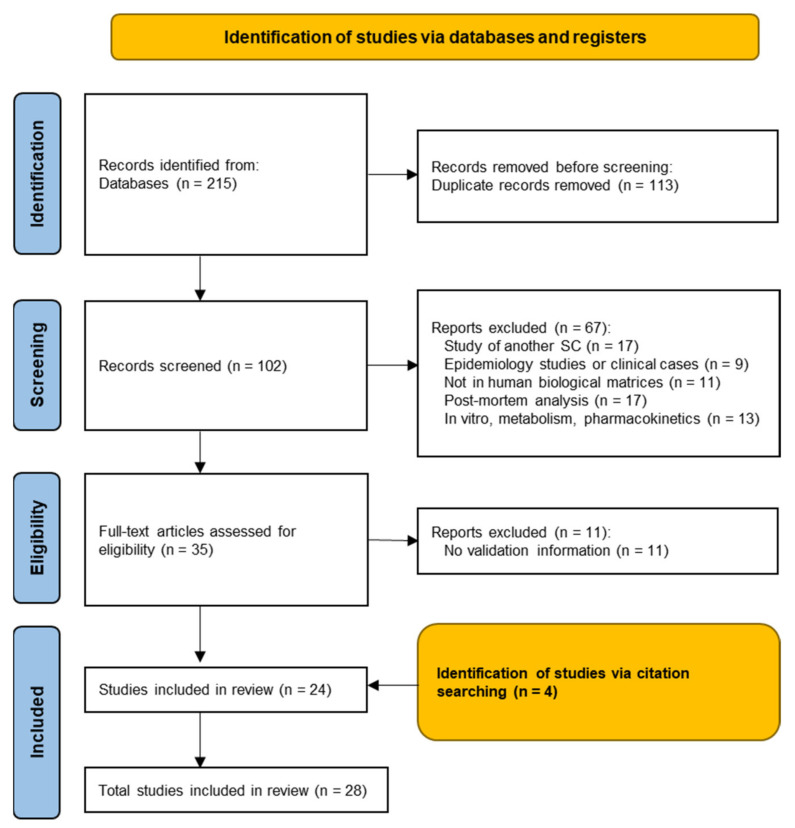
PRISMA flowchart for the selected studies.

**Table 1 biology-11-00796-t001:** Nomenclature, structure, and characteristics of the four synthetic cannabinoids included in this review.

	AB-CHMINACA	MDMB-CHMICA	5F-MDMB-PINACA *	ADB-CHMINACA
IUPAC name	N-(1-Amino-3-methyl-1-oxobutan-2-yl)-1-(cyclohexylmethyl)-1H-indazole-3-carboxamide	Methyl 2-({[1-(cyclohexylmethyl)-1H-indol-3-yl] carbonyl}amino)-3,3-dimethylbutanoate	Methyl 2-{[1-(5-fluoropentyl)-1Hindazole-3-carbonyl]amino}-3,3-dimethylbutanoate	N-(1-Amino-3,3-dimethyl-1-oxobutan-2-yl)-1-(cyclohexylmethyl)-1H-indazole-3-carboxamide
Street names **	Aromatic Pot Pourri, Jamaican Gold Supreme, Bonzai Citrus, Blaze, Bubblegum, Manga Xtreme, Matt Hardener, Aura mystic Bulc	Godfather, CUSHCottonCandy, KUSHSecondGereration,KUSHherbal incense, Ninja, Sirius, SKIHIGH, CRITICAL haze	ANNIHILATION, BLACK MAMBA ULTRA, Blueberry Blitz, CHERRY BOMB, Dutchy, EXODUS FORMULA 6-A, Sky High, Spike 99 ULTRA, and Vanilla Ice	ADB-CHMINACA, MAB-CHMINACA
Molecular formula	C_20_H_28_N_4_O_2_	C_23_H_32_N_2_O_3_	C_20_H_28_FN_3_O_3_	C_21_H_30_N_4_O_2_
Molecular weight(g/mol)	356.5	384.5	377.5	370.5
Structure	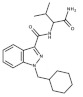 Linked group: methyl amino butanone (AB)Tail: cyclohexylmethyl (CHM)Core: indazole (INA)Linker: carboxamide (CA)	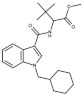 Linked group: methyldimethylbutanoate (MDMB)Tail: cyclohexylmethyl (CHM)Core: indole (I)Linker: carboxamide (CA)	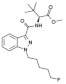 Linked group: dimethyl methyl butanoate (MDMB)Tail: pentyl (P), with a fluoro moiety at the position 5Core: indazole (INA)Linker: carboxamide (CA)	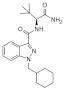 Linked group: dimethylaminobutanone (ADB)Tail: cyclohexylmethyl tail (CHM)Core: indazole (INA)Linker: carboxamide (CA)
Pharmacology and toxicology	Full and partial agonist of the CB1 and CB2 receptors, respectively;11–58 times more potent than THC [33]	Potent and full agonist of the CB1 receptor and agonist at the CB2receptor;400 times more potent than THC, two times more potent than AB-CHMINACA [34]	Potent full agonist at the CB1 and CB2 receptor;289 times more potent than THC, 17 times more potent than MDMB-CHMICA [35]	Potent and full agonist of the CB1 receptor and agonist at the CB2receptor;270 times more potent than THC [36]
Detection and evaluation by the EMCDDA	First detected: February 2014 (Riga, Latvia)First notified to the EMCDDA:April 2014Risk-assessed by the EMCDDA in 2017	First detected: August 2014 (Hungary)First notified to the EMCDDA:September 2014Risk-assessed by the EMCDDA in 2016	First detected: September 2014 (Hungary)First notified to the EMCDDA: January 2015Risk-assessed by the EMCDDA in 2017	First detected: Synthesis of ADB-CHMINACA was first described in a 2009 patent by PfizerFirst notified to the EMCDDA: September 2014Risk-assessed by the EMCDDA in 2017
Psychological and behavioral effects	Duration of the effect: 1–2 h after smokingEffects: cannabis- and THC-like (relaxation, confusion, anxiety…); psychotic episodes and aggressive behaviors have also been reported	Duration of the effect: 120 min after smokingEffects: more pronounced in comparison to cannabis; most common paranoia, euphoria, visual hallucinations, and anxiety	Duration of the effect: 1–2 h after smoking. Effects lasting more than 10 h have been describedEffects: cannabis- and THC-like (relaxation, confusion, anxiety); psychotic episodes and aggressive behaviors have also been reported.	Effects: cannabis- and THC-like (relaxation, confusion, anxiety…); psychotic episodes and aggressive behaviors have also been reported
Some analytical identification techniques used based on the EMCDDA	GC-MS; FTIR-ATR; HPLC-TOF; NMR; LC-MS; UV-VIS; LRMS; HRMS; DART-MS [37]	NMR and HPLC-DAD for quantification in products. LC-MS/MS for detection in biological samples [38]	HPLC-DAD for quantification in products. LC-MS/MS for detection in biological samples [39]	In products: GC-MS, LC-QqQ-MS/MS, GC-TOF-MS, GC-EI-MS, NMR, LC-MS/MS, IR, UVIn biological samples: LC-QqQ-MS/MS, LC-MS/MS, LC-QTRAP-MS/MS, GC-MS, LC-QTOF-MS [40]

* Also known as 5F-ADB; **: due to the frequently changing cannabinoid content of the product, it is possible that the name does not match current SC content. Abbreviations: DART-MS: direct analysis in real time; FTIR-ATR: Fourier transform infrared spectroscopy attenuated total reflectance; GC-EI-MS: gas chromatography/electron ionization mass spectrometry; GC-MS: gas chromatography-mass spectrometry; GC-TOF-MS: gas chromatography coupled to time-of-flight mass spectrometry; HPLC-DAD: high-performance liquid chromatography-diode array detector; HPLC-TOF: high-performance liquid chromatography time-of-flight; HRMS: high-resolution mass spectrometry; IR: infrared spectroscopy; LC-MS: liquid chromatography-mass spectrometry; LC-MS/MS: liquid chromatography with tandem mass spectrometry; LC-QqQ-MS/MS: liquid chromatography triple quadrupole tandem mass spectrometry; LC-QTOF-MS: liquid chromatography quadrupole time-of-flight mass spectrometry; LC-QTRAP-MS/MS: ultra-high-performance liquid chromatography coupled with QTRAP mass spectrometry; LRMS: low-resolution mass spectrometry; NMR: nuclear magnetic resonance spectroscopy; UV-VIS: ultraviolet-visible spectroscopy.

**Table 2 biology-11-00796-t002:** Summary of the analytical methods for the identification of synthetic cannabinoids in biological matrices.

Matrix	Study/Country	Qual./ Quant.	Analyzed SCs	Sample Preparation	Detection Method	Type and Details of Samples	Study Limitations as Reported by the Authors
HAIR	Cho et al., 2020 [45]/(South Korea)	Quant.	18 SCs and 41 of their metabolites, including: AB-CHMINACA and AB-CHMINACA M1A, M2 and M4	Washed with methanol and cut finely (1 mm) and dried at room temperature. Incubation with 2 mL methanol at 38 °C, evaporation under nitrogen gas, and filtration	LC-MS/MS	Hair samples from 43 individuals who were suspected of using SCs. (January 2016–December 2018)	Not reported
	Sim et al., 2017 [46] (South Korea)	Quant.	AB-CHMINACA and its six metabolites: M2, M4, M3A, M5A, M6, and M7	Washed with methanol and distilled water, through-flow dried, and cut into 1–2 mm pieces. Incubation with 2 mL of methanol at 38 °C, evaporation under nitrogen gas at 45 °C and filtration	LC-MS/MS	122 hair samples from suspects who were suspected of using SCs and had been arrested by the police. (November 2015–November 2016)	Not reported
	Franz et al., 2016 [47] (Germany)	Qual. and semi-quant. for parent compounds Qual. for metabolites	AB-CHMINACA and its metabolite AB-CHMINACA M2	Washed by shaking in deionized water, acetone, and petroleum ether. Dried, cut into 1 mm pieces, and extracted by ultrasonication. Dried under nitrogen at 40 °C	LC-MS/MS	Hair sample collected from a 16-year-old female withdrawal patient for abstinence control	Findings in the hair segments do not correlate with use of the drug in the period at which the corresponding hair segments had grown The distribution of the detected compounds is suggestive of incorporation via sebum and sweat
	Franz et al., 2018 [20] (Germany)	Qual. and semi-quant.	AB-CHMINACA, ADB-CHMINACA, MDMB-CHMICA, and 5F-MDMB-PINACA	Washed by shaking in deionized water, acetone, and petroleum ether. Dried, cut into 1 mm pieces, and extracted by ultrasonication. Dried under nitrogen at 40 °C	LC-MS/MS	294 hair samples (drug abstinence testing) (January 2012–December 2016)	High matrix effects The exact LODs were not determined individually (estimated to be around one order of magnitude lower for most analytes compared to the LLOQs).
ORAL FLUID	Williams et al., 2019 [48] (Australia)	Quant.	19 SCs, including AB-CHMINACA	Protein precipitation	LC-MS/MS	12 authentic samples submitted for routine testing in which no cannabinoids were detected	Lack of confirmed positive samples, lack of an external quality assurance program
	Cooman et al., 2020 [49] (USA and Brazil)	Quant.	24 SCs and cathionine derivatives, including AB-CHMINACA	SPE	LC-MS/MS	Blind study that included 10 OF samples from volunteers, prepared with varying concentrations of analytes	LLOQ bias of 33.6% for AB-CHMINACA AB-CHMINACA values > 20% greater than the highest calibrator due to matrix and ion suppression/enhancement effects or to samples being prepared at higher concentrations than expected.
	Sorribes-Soriano et al., 2021 [50] (Spain)	Quant.	5 SCs, including 5F-MDMB-PINACA	SPE by MEPS	GC-MS	Pool of 15 saliva samples from different volunteers spiked with a synthetic cannabinoid at 125 and 250 μg/L	Not reported
	Denia et al., 2022 [51] (Spain)	Quant.	5F-MDMB-PINACA	Extraction by chloroform mixture and phase separation by centrifugation	GC-IMS	Pool of OF samples from five non-consumer volunteers with known concentrations of the added SCs	Not reported
BLOOD	Peterson and Couper, 2015 [52] (USA)	Quant.	40 SCs, including AB-CHMINACA	LLE	LC-MS/MS	6815 blood samples from suspected impaired driving cases	Tests were no uniformity in the performed tests among all cases, as the number of compounds screened increased over the year
	Tynon et al., 2017 [53] (USA)	Qual.	34 SCs, including AB-CHMINACA and ADB-CHMINACA	LLE using MTBE	LC-MS/MS	1497 blood samples from forensic investigations, including postmortem examinations and driving impairment cases (March 2015–December 2015)	AB-CHMINACA and ABD-CHMINACA did not meet the requirements for quantitative confirmation
	Adamowicz and Gieroń, 2016 [31] (Poland)	Quant.	ADB-CHMINACA	Protein precipitation	LC-MS/MS	Blood samples from four adolescents who had smoked a substance labeled “AM-2201”	Not reported
	Adamowicz, 2016 [32] (Poland)	Quant.	MDMB-CHMICA	Protein precipitation	LC-MS/MS	Antemortem and postmortem blood sample of a 25-year-old male with fatal intoxication due to SC abuse	Not reported
	Hess et al., 2017 [54] (Germany)	Qual. and quant.	93 SCs, including AB-CHMINACA, MDMB-CHMICA, 5F-MDMB-PINACA, and ADB-CHMINACA	LLE	LC-MS/MS	189 blood samples from suspected drugged individuals while diving (January 2013–November 2015)	When applied to real case samples, quantification ranges of many of the compounds were lower than LLOQ.
	Seywright et al., 2016 [55] (U.K.)	Quant.	MDMB-CHMICA	LLE	LC-MS/MS	26 cases suspected of having consumed SC at the Emergency Department of Glasgow RoyalInfirmary	Small number of cases No metabolite screening because no reference standards were available. This may have increased the detection window
	Bäckberg et al., 2017 [34] (Sweden)	Quant.	MDMB-CHMICA	Protein precipitation	LC-HRMS	Eight intoxication cases involving MDMB-CHMICA from the pool of samples from the STRIDA project (2014–2015)	Small sample size Possible interferences by other psychoactive substances Difficulty in the identification of MDMB-CHMICA due to the unknown stability of the compound and inter-individual variability of drug metabolism
	Grapp et al., 2018 [56] (Germany)	Quant.	950 compounds (185 drugs and metabolites), including AB-CHMINACA and MDMB-CHMICA	LLE	LC-QTOF-MS	Analysis 247 drug-positive serum and 12 post mortem femoral blood samples submitted by the police of Lower Saxony with the request for drug analysis	For the correct identification of compounds, data verification by a toxicologist was needed.
	Saito et al., 2020 [57] (Japan)	Qual.	47 SCs, including AB-CHMINACA	SPDE	LC/TOF-MS	Blood samples (no additional specifications)	Not reported
	Krotulski et al., 2020 [58] (USA)	Qual.	247 SCs, including AB-CHMINACA, 5F-MDMB-PINACA, MDMB-CHMICA, ADB-CHMINACA in blood and AB-CHMINACA M2, 5F-MDMB-PINACA M20, ADB-CHMINACA M2 in urine	LLE	LC-QTOF-MS	200 authentic blood samples suspected of containing synthetic cannabinoids; 104 were compared against the results provided by the toxicology laboratory (June 2018)	Not reported
	Ong et al., 2020 [59] (New Zealand)	Qual. and semi-quant.	29 SCs and metabolites including 5F-MDMB-PINACA, 5F-MDMB-PINACA M20, AB-CHMNACA, AB-CHMINACA M1A, MDMB-CHMICA and MDMB-CHMICA M30	SLE	LC-MS/MS	564 authentic human blood samples: Postmortem examinations (*n* = 180); criminal cases (*n* = 8); impaired drivers (*n* = 343); emergency department admissions (*n* = 19); psychiatric care patients (*n* = 14)	The validation evaluated an inadequate distribution of concentration points; therefore, exact quantitative values were not reported
URINE	Franz et al., 2017 [60] (Germany)	Qual. and quant.	Qual.: 130 metabolites from 45 SCs Quant.: 31 metabolites from 14 SCs Including metabolites from AB-CHMINACA, ADB-CHMINACA, and MDMB-CHMICA	SPE	Immunoassay confirmed by LC-MS/MS	Study A: 549 urine samples from a regular drug screening (October–November 2014) Study B: 100 negative and 100 positive urine samples included in the study from a regular drug screening (January–June 2015)	LC-MS/MS was not fully validated for the assessed analytes (reference standards not commercially available): a similar fragmentation pattern of a parent compound was assumed. A limited number of positive samples was analyzed because samples positive for metabolites of more than one SC were excluded
	Dybowski et al., 2021 [35] (Poland)	Qual. and quant.	5F-MDMB-PINACA and its degradation products	QuEChERS extraction (combination of LLE + d-SPE)	GC-MS/MS	Urine samples from volunteers spiked with 5F-MDMB-PINACA	Very low recovery (<30%) of the drug from alkaline urine.
	Kakehashi et al., 2020 [61] (Japan)	Quant.	AB-CHMINACA, 5F-MDMB-PINACA	LLE	LC-MS/MS	27 urine samples from drivers involved in car crashes allegedly under the influence of SCs (2011–2014)	Quantification was impossible for some urine specimens due to insufficient sample volume
	Institóris et al., 2017 [62] (Hungary)	Qual. and quant.	100 SCs, including AB-CHMINACA ADB-CHMINACA MDMB-CHMICA	Enzymatic hydrolysis and SLE	UHPLC-MS/MS	271 urine samples from drivers suspected to have used DUID (2014–2015)	Incomplete clinical data collection Unanalyzed substances may have affected the half-life of the analyzed ones
	Franz et al., 2017 [63] (Germany)	Qual.	MDMB-CHMICA and the M25 and M30 metabolites	LLE	LC-MS/MS	5717 authentic urine samples in controls of abstinence control (October 2014–November 2015)	Exact structure of some metabolites is unknown (impossible by NMRS) Phase II metabolites could not be covered because of the glucuronide cleavage step in sample preparation. Polymorphisms in CYP450 isoenzymes were not studied but could influence individual metabolic profiles.
	Yeter and Ozturk, 2019 [64] (Turkey)	Quant.	5F-MDMB-PINACA and the M20 metabolite	SPE +/− enzymatic hydrolysis	LC-HRMS	30 samples chosen from screening of 8235 authentic urine samples from drug use suspects (January 2017–June 2018)	Not reported
	Gundersen et al., 2019 [25] (Norway)	Qual. and quant.	35 SC metabolites, including AB-CHMINACA M1A and AB-CHMINACA M4.	SPE	UHPLC-QTOF-MS	1000 urine samples from individuals in drug withdrawal programs (throughout 2014 and the first half of January 2015)	Due to matrix effects, low recoveries and linearities, and lack of isotopically labeled internal standards, the method should be considered semi-quantitative for AB-CHMINACA M1A and AB-CHMINACA 3-carboxyindazole
	Tyndall et al., 2015 [65] (USA)	Quant.	50 SCs and metabolites + formula matches for 157 other SC parent compounds and 13 predicted AB-CHMINACA metabolites (including M2, M6, M11) Other drugs of abuse	Dilute and shoot method	LC-QTOF-MS	21 urine samples from patients presenting to the emergency department with a documented suspicion of SCs use (May–June 2014)	Not reported
	Cannaert et al., 2017 [66] (Belgium and Germany)	Quant.	Four SCs, including AB-CHMINACA and its metabolites M1A, M1B, M2, M3A. ADB-CHMINACA and its metabolites M1, M2, M3	LLE	LC-MS/MS (+CB Reporter Assays)	74 authentic urine samples from suspected SC users	High concentrations of metabolites in urine are required for detection. False negative results for low concentrations of AB-CHMINACA Useful as a pre-screening tool but requires other analytical techniques for confirmation

Abbreviations: AMR: analytical measurement range, CB: cannabinoid receptor, D-SPE: dispersive solid-phase extraction, GC-MS: gas chromatography-mass spectrometry, GC-IMS: gas chromatography-ion mobility spectrometry, LC-HRMS: liquid chromatography high-resolution mass spectrometry, LC-MS/MS: liquid chromatography with tandem mass spectrometry, LC-QTOF-MS: liquid chromatography-quadrupole time-of-flight mass spectrometry, LC-TOF-MS: liquid chromatography time-of-flight mass spectrometry, LLE: liquid-liquid extraction, MEPS: semi-automated microextraction by packed sorbent, MTBE: methyl tertiary-butyl ether, OF: oral fluid. QuEChERS: acronym for quick, easy, cheap, effective, rugged, and safe, SCs: synthetic cannabinoids, SLE: supported liquid extraction, SPE: solid-phase extraction. SPDE: solid-phase dispersive extraction, UHPLC-MS/MS: ultra-high-performance liquid chromatography with tandem mass spectrometry, UHPLC-QTOF-MS: ultra-high-performance liquid chromatography-quadrupole time-of-flight mass spectrometry, Quant.: quantitative, Qual: qualitative.

**Table 3 biology-11-00796-t003:** Validation parameters measured in the analytical methods used for the identification of synthetic cannabinoids.

Studies	LOD	LLOQ	Accuracy	Linearity	Matrix Effect	Precision	Process Efficiency	Recovery	Selectivity	Sensitivity	Specificity	Stability	Carryover
Cho et al., 2020 [45]	✓	✓	✓	✓	✓	✓	✓	✓	✓	✓	✓	✓	
Sim et al., 2017 [46]	✓	✓	✓	✓	✓	✓	✓	✓	✓			✓	
Franz et al., 2016 [47]	✓	✓	✓		✓	✓			✓			✓	
Franz et al., 2018 [20]	✓		✓	✓	✓	✓	✓	✓	✓				
Williams et al., 2019 [48]	✓	✓	✓	✓		✓			✓				✓
Cooman et al., 2020 [49]		✓	✓	✓	✓	✓	✓	✓	✓			✓	✓
Sorribes-Soriano et al., 2021 [50]	✓	✓		✓		✓		✓	✓	✓			
Denia et al., (2022) [51]	✓			✓		✓		✓					✓
Peterson and Couper, 2015 [52]	✓	✓	✓	✓	✓								✓
Tynon et al., 2017 [53]					✓	✓				✓	✓	✓	✓
Adamowicz and Gieroń, 2016 [31]	✓	✓	✓	✓	✓	✓		✓	✓		✓		
Adamowicz, 2016 [32]	✓	✓	✓	✓	✓	✓		✓	✓		✓		
Hess et al., 2017 [54]	✓	✓	✓	✓	✓	✓			✓		✓		
Seywright et al., 2016 [55]	✓	✓	✓	✓	✓	✓	✓		✓				
Grapp et al., 2018 [56]	✓				✓			✓			✓		
Saito et al., 2020 [57]	✓					✓		✓					
Krotulski et al., 2020 [58]	✓		✓			✓			✓	✓		✓	✓
Ong et al., 2020 [59]	✓		✓	✓	✓	✓	✓		✓	✓		✓	✓
Franz et al., 2017 [60]	✓	✓	✓	✓	✓	✓		✓	✓	✓	✓		
Dybowski et al., 2021 [35]	✓	✓	✓	✓	✓	✓		✓	✓	✓	✓		
Kakehashi et al., 2020 [61]	✓		✓	✓		✓		✓					
Institóris et al., 2017 [62]	✓	✓	✓	✓		✓							
Franz et al., 2017 [63]	✓	✓	✓		✓	✓			✓			✓	
Yeter and Ozturk 2019 [64]	✓	✓	✓	✓	✓	✓		✓				✓	
Gundersen et al., 2019 [25]		✓	✓	✓	✓	✓		✓	✓		✓	✓	✓
Cannaert et al., 2017 [66]	✓	✓	✓	✓	✓	✓		✓					

The check symbol indicates the validation parameters measured in each study. Abbreviations: LLOQ: lower limit of quantification. LOD: limit of detection.

## Data Availability

Not applicable.

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
