# Peer review of "Detection of the Synthetic Cannabinoids AB-CHMINACA, ADB-CHMINACA, MDMB-CHMICA, and 5F-MDMB-PINACA in Biological Matrices: A Systematic Review"

_biology, 2022, doi:10.3390/biology11050796_

Round 1
Reviewer 1 Report
The manuscript is well written and comprehensively covers the literature. However, the usefulness to the readers seems limited, especially because it contains several well-known concepts (see, for example, the entire Discussion paragraph), and it includes a very small panel of analytes. A review should instead facilitate researchers to have access to information on an extended topic.
Headline of Table 3 is not readable
Author Response
The manuscript is well written and comprehensively covers the literature. However, the usefulness to the readers seems limited, especially because it contains several well-known concepts (see, for example, the entire Discussion paragraph), and it includes a very small panel of analytes. A review should instead facilitate researchers to have access to information on an extended topic.
Thank you for your suggestions. We are aware of the limitations of the study, and in fact we point this out at the end of the discussion. To date, more than 280 SCs have been reported worldwide, so we decided to choose the most relevant SCs due to their novelty in the market (1,2). In addition, there have also been cases of other drugs adulterated with these SCs (3). Our intention, in addition to reviewing the most used techniques for their detection, is to present the latest studies on these SCs in the main biological matrices. To date, there have been
reviews of SCs focused particularly on their effect but, according to the PROSPERO database, there are no systematic reviews focused on the detection of these SCs based on the matrix and the method used. Knowledge on the subject is scarce, despite searching for information on the 4 SCs and their detection in 4 different biological matrices, so our aim has been to collect all
that information published in recent years and unify it, to clarify what is known to date. In addition, we wanted to draw attention to the need to use other matrices, such as fingernails or toenails, and to present emerging identification techniques, which do not require previous standards, such as HRMS. We hope that this review will help professionals in this field to explore new techniques and matrices, not so widely used to date, and open the door to their further understanding.
Likewise, we have made substantial changes to the manuscript. First, we have sent the manuscript to a professional translator, so the language quality of the whole paper has improved substantially. In addition, we have rearranged some paragraphs so that the order of the matrices we are talking about (urine, hair, blood and saliva) is always the same throughout the text. We have expanded the information in tables 1 and 2 and we have also made them more visually attractive, as well as table 3, where we have changed the crosses for check symbols to make it more understandable.
Headline of Table 3 is not readable. We have changed the table to do it more readable.
1- Hermanns-Clausen M, Müller D, Kithinji J, Angerer V, Franz F, Eyer F, Neurath H, Liebetrau G, Auwärter V. Acute side effects after consumption of the new synthetic cannabinoids AB-CHMINACA and MDMB-CHMICA. Clin Toxicol (Phila). 2018 Jun;56(6):404-411. doi: 10.1080/15563650.2017.1393082. Epub 2017 Oct 26. PMID: 29072524.
2- Carlier J, Diao X, Sempio C, Huestis MA. Identification of New Synthetic Cannabinoid. ADB-CHMINACA (MAB-CHMINACA) Metabolites in Human Hepatocytes. AAPS J. 2017 Mar;19(2):568-577. doi: 10.1208/s12248-016-0037-5. Epub 2017 Jan 9. PMID:
28070717.
3- Ershad M, Dela Cruz M, Mostafa A, Khalid MM, Arnold R, Hamilton R. Heroin
Adulterated with the Novel Synthetic Cannabinoid, 5F-MDMB-PINACA: A Case Series. Clin Pract Cases Emerg Med. 2020;4(2):121-125. Published 2020 Apr 23.
doi:10.5811/cpcem.2020.2.45060
Reviewer 2 Report
The paper could be very interesting but it has a very nebulous and confusing organization. There is a lack of logical order in the structure of the paper which is very difficult to read. I suggest to the Authors to privilege a literature review either in relation to the analytical methods (screening, GC and/or LC/MS or tandem mass spectrometry, HRMS) or to the biological matrices under investigation.
- In the Simple Summary the Authors cite the biological matrices in this sequence: urine, hair, blood and oral fluid but in the results section the order changes. To facilitate the readers I suggest to always follow the same order.
- In Table 1 for 5F-MDMB-PINACA and ADB-CHMINACA the Authors use also other names not reported in the main text. It would be better to use no other names to avoid confusion.
- Report in Table 1 the row names for each molecule (Structure, Action, EMCDDA detection.....) to better understand the table.
- Line 194 and 206: "Table" is in italic.
- Table 2: the sentence NOT APPLIED TO REAL CASES is sometimes reported in the column of "STUDY LIMITATIONS REPORTED BY AUTHORS".
- Table 2: check the order of the biological matrices and choose only one in according to the main text.
- Table 3 is not comprehensible. Please change the settings of the validation parameters.
- Results - 3.2 Oral fluid: as for hair paragraph, list the disadvantages of using saliva.
- 3.2 Oral fluid: it's the first time reporting MEPS, write the name of the technique in full.
- Discussion: choose the sequence of reportin biological matrices.
I suggest the Authors to proofread the paper by a native English speaker.
Author Response
Dear reviewer,
The Response to your suggestions and recommendations is in the word file attached.
Thanks for your comments
